# Analysis of Potential Genes and Economic Parameters Associated with Growth and Heat Tolerance in Sheep (*Ovis aries*)

**DOI:** 10.3390/ani13030353

**Published:** 2023-01-19

**Authors:** Samer Ibrahim, Mona Al-Sharif, Fawzy Younis, Ahmed Ateya, Mohamed Abdo, Liana Fericean

**Affiliations:** 1Department of Animal Husbandry and Animal Wealth Development, Faculty of Veterinary Medicine, Mansoura University, Mansoura 35516, Egypt; 2Department of Biology, College of Science, University of Jeddah, Jeddah 23218, Saudi Arabia; 3Animal and Poultry Physiology Department, Animal and Poultry Division, Desert Research Center, Cairo 11753, Egypt; 4Department of Animal Histology and Anatomy, School of Veterinary Medicine, Badr University in Cairo (BUC), Cairo 11829, Egypt; 5Department of Anatomy and Embryology, Faculty of Veterinary Medicine, University of Sadat, Sadat City 32897, Egypt; 6Department of Biology and Plant Protection, Faculty of Agricultural Sciences, University of Life Sciences King Michael I, 300645 Timisoara, Romania

**Keywords:** candidate gene, sheep, growth, heat tolerance, economic evaluation

## Abstract

**Simple Summary:**

Understanding the genetic basis of animal growth and heat tolerance is essential if genetic enhancement technologies are employed to promote these traits. This study examined potential genes and economic factors that might be associated with growth and heat tolerance in two sheep breeds. In total, 120 male lambs of the Barki and Aboudeleik breeds, 60 of each breed, were used in this investigation. Lambs from Barki have much better growth traits than those from Aboudeleik. PCR-DNA sequencing of the growth and heat tolerance genes revealed SNPs in the nucleotide sequences between Barki and Aboudeleik lambs. The discovered SNPs were found to significantly associate with growth and heat tolerance traits. Lambs from Barki had a better economy than those from Aboudeleik.

**Abstract:**

This study explored the potential genes and economic factors that might be associated with growth and heat tolerance in two sheep breeds. Data on growth performance from the third month to six months of age were obtained based on records. In comparison to Aboudeleik lambs, Barki lambs developed considerably greater starting body weight, final body weight, final body weight gain, daily weight gain, and percentage increase in BW/month. Single nucleotide polymorphisms (SNPs) were found between lambs of the two breeds using PCR-DNA sequencing of *CAST*, *LEP*, *MYLK4*, *MEF2B*, *STAT5A*, *TRPV1*, *HSP90AB1*, *HSPB6*, *HSF1*, *ST1P1*, and *ATP1A1* genes. Lambs from each breed were divided into groups based on detected SNPs in genes related to growth. The least squares means of the differentiated groups revealed a significant correlation of detected SNPs with growth and heat tolerance attributes (*p* ≤ 0.05). Barki lambs elicited greater total variable costs, total costs, total return, and net return values. The Barki sheep provided the best economic efficiency value when comparing the percentage difference between net profit and economic efficiency. Together with economic considerations, SNPs found may be used as proxies for marker-assisted selection of the best breed of sheep for traits related to growth and heat tolerance.

## 1. Introduction

In dry areas, small ruminants are necessary for providing meat and meat products. The sheep (*Ovis aries*) is among the most important domestic animals in terms of economy, culture, and society [1]. In terms of the Egyptian agricultural industry, sheep farming provides farmers with a great source of income and a significant source of meat and milk [2]. Sheep make up around 6.4% of the nation’s overall production of red meat in Egypt. In the past five years, Egypt’s sheep population has grown quickly, reaching almost 5.7 million animals in 2017 [3]. With a population of 470,000 individuals (8.5% of the total Egyptian sheep population), Barki sheep are distributed over Egypt’s northwestern coastal zone (NWCZ). A half -million -head population of the most significant breed of sheep, the Barki originated in the Libyan city of Burka and was raised along the northern coast [4]. When compared to other indigenous breeds, it is one of the fat-tailed sheep breeds with the highest development potential [5]. Because Barki sheep are well-adapted to harsh desert situations, including a lack of fodder and high ambient temperatures, they can produce a significant amount of meat, wool, and milk in these circumstances [4]. Aboudeleik sheep are recognized as one of the essential breeds in the triangle for producing milk and meat because of their fortitude in the face of adversity [6]. Their typical body weight is 45 kg, and short hair covers a shallow body with lengthy legs [6]. Both sexes have lengthy heads and necks, frequently with a dewlap. The ears are nonexistent or small, with a backward downward angle, and the tail is cylindrical and long, extending below the hocks. The color is black or brown with cream [7].

Growth -related traits in sheep are of strong economic interest in Egypt [8]. Climate is a word used to describe the general or long-term weather patterns in a region. It is made up of a number of variables, including temperature, humidity, rainfall, air movement, radiation, barometric pressure, and ionization [9]. High ambient temperatures are the main factor limiting animal productivity in tropical and subtropical locations [10,11]. Reduced body weight, average daily gain (ADG), growth rate, and total body solids are all indicators of reduced production in sheep exposed to high temperatures [12,13]; therefore, sheep farmers suffer huge financial losses. However, as global warming progresses, total economic losses resulting from animals suffering heat stress have now exceeded $2.4 billion yearly in the US [14]. Consequently, reducing the effects of heat stress is crucial for animals. The physico-biochemical characteristics of sheep are similarly affected by heat stress [15,16].

Animals frequently show phenotypic variation in terms of growth and heat tolerance [17]. The genes/alleles underlying these traits can be found by looking at this variability at the DNA level, and this information can then be used to choose the best animals [18,19]. The use of genomic technology has advanced molecular genetics, creating exciting opportunities for the discovery of functional genes. Genome sequencing projects have allowed the discovery of thousands of single nucleotide polymorphisms (SNPs) across the genome of different livestock species. These genetic markers, or SNPs, can be used to find genetic variability underlying features in livestock animals that are economically valuable and to better understand how genetic variants relate to different phenotypes [20]. Nowadays, the development of accessible sequencing technologies and large data analysis software has made significant contributions to our understanding of the genetic pathways and genetic foundation of various phenotypic features in different species, including sheep [21]. As a result, genetically based improvement programs should be designed with the aim of facilitating the selection of breeding animals, a selection which will genuinely increase qualities related to growth and heat tolerance.

The immune system is impacted by heat stress in a complicated and changing way [22]. Exposure to a heat -stress environment modifies physiological traits, which may change how much cytokines are circulated. In order to control how the body responds to stimuli, the hypothalamic-pituitary-adrenal (HPA) axis and the sympathetic adrenal medullary system (SAM) are crucial [23]. Catecholamines are thought to have an inflammatory effect, whereas cortisol is produced as a result of stimulation of the HPA axis and is linked to a reduction of the immune system in animals [24]. It was determined that cows under a heat -stress environment had greater circulating levels of interleukin-1ß (IL-1ß) and interleukin-6 (IL-6) than cows under a non- or mildly stressful environment [25]. The latter raises the likelihood that the heat -stress environment may have caused an inflammatory condition [26,27].

Previous research examined the growth performance of sheep based on phenotypic characteristics [28,29]. However, no studies have previously examined sheep growth traits from an economic standpoint and by taking into account a candidate gene. Many studies on heat stress in livestock have primarily focused on temperature and relative humidity [30,31]. The molecular basis of sheep heat tolerance is not well understood [32]. Therefore, it is necessary to do additional research involving functional verification by using genetic approaches that better incorporate statistical analyses.

The aim of this study was to analyze candidate genes and economic parameters associated with the growth performance and heat tolerance in the Barki and Aboudeleik breeds of sheep. Another goal was to investigate the association of nucleotide sequence variants in growth -related genes with growth performance and economic parameters. In order to understand the genetic variance in heat adaptability in the two breeds, the ultimate aim was to explore the effect of polymorphism in heat tolerance genes on physico-chemical parameters.

## 2. Materials and Methods

### 2.1. Ethics Statement

Under the direction of the Faculty of Veterinary Medicine at the University of Sadat City, all animal handling protocols, sample collection, and sample disposal were completed in accordance with IACUC rules (Ethical approval number: VUSC-006-1-22).

### 2.2. Animals and Sampling

A total of 120 male lambs, 60 of each breed (Barki and Aboudeleik), were used in this investigation. Lambs from Barki and Aboudeleik were 6 months old, with average final body weights of 41.06 and 36.57 kg, respectively. Based on the station records, data for growth performance parameters were collected from the third month of age until the six-month. At three months of age, lambs from Barki and Aboudeleik had average initial body weights of 27.65 kg and 26.92 kg, respectively. Animals were kept at the Desert Research Center, Ministry of Agriculture, and Land Reclamation’s Ras Sudr Research Station in the South Sinai Governorate. All animals were housed in identical housing circumstances throughout the duration of the trial and had no prior history of metabolic illnesses. Throughout the course of the experiment, animals were kept in semi-open enclosures and spent roughly 6 h each day grazing on open pastures. In accordance with National Research Council (NRC) (2007) [33] recommendations, the basal diet was developed to meet the nutritional requirements of the lambs and provide them with the nutrients and energy they required (Table 1).

Ten milliliters of blood were extracted from each lamb’s jugular veins. The samples were collected into a vacutainer tube containing anticoagulant (EDTA or sodium fluoride) and without anticoagulant to yield whole blood or serum, respectively. The EDTA blood was used for DNA extraction and estimation of red blood cells (RBCs) and mean corpuscular volume. While those in plain tubes were kept overnight at room temperature and centrifuged at 3000 r/min for 15 min. Only clear sera were collected then aliquoted and kept frozen at −20 °C for subsequent cytokine analysis.

### 2.3. Productive Parameters

#### Body Weight Gain (BWG) and Daily Gain

Body weight gain of growing lambs was calculated as the difference between two successive weights each week. The final body weight gain is also calculated as the difference between the initial body weight and the final body weight. In addition, the daily gain is calculated by dividing the final BWG by 90 days (the period of study) [34].
BWG = BW1 − BW2 
$$\mathrm{Daily}\;\mathrm{WG}\;=\frac{\mathrm{final}\;\mathrm{BWG}}{90\;\mathrm{day}}$$

### 2.4. DNA Extraction and Polymerase Chain Reaction (PCR)

Using the Gene JET whole blood genomic DNA extraction kit (Thermo Scientific, Lithuania) and following the manufacturer’s instructions, genomic DNA was extracted from whole blood. The good quality, purity and concentration of the DNA were screened by Nanodrop (Uv-Vis spectrophotometer Q5000/USA). The following coding site (CDS)-based growth and heat stress gene segments were amplified using PCR: *CAST* = calpastatin; *LEP* = leptin; *MYLK4* = myosin light chain kinase family member 4; *MEF2B* = myocyte enhancer factor 2B; *STAT5A* = signal transducer and activator of transcription 5A; *TRPV1* = transient receptor potential cation channel subfamily V member 1; *HSP90AB1* = heat shock protein 90 alpha family class B member 1; *HSPB6* = heat shock protein family B (small) member 6; *HSF1* = heat shock transcription factor 1; *ST1P1* = stress -induced phosphoprotein 1 and *ATP1A1* = ATPase Na+/K+ transporting subunit Alpha 1. The primer sequences were designed in accordance with the *Ovis aries* sequence that was published in PubMed (https://www.ncbi.nlm.nih.gov/nuccore/?term=ovis+aries, accessed on 8 May 2022) using basal local alignment search tool (BLAST). Table 2 lists the primers and accession numbers for genes used in the amplification.

The polymerase chain reaction mixture was run in a thermal cycler with a final volume of 100 μL. Each reaction volume contained 1 μL of each primer, 25 μL of PCR master mix (Jena Bioscience, Jena, Germany), 5 μL DNA, and 68 μL H2O (d.d. water). The first denaturation temperature of 94 °C was applied to the reaction mixture for 8 min. The PCR was carried out for 30 cycles of denaturation at 94 °C for 1 min, annealing at temperatures (as stated in Table 2) for 45 s each, extension at 72 °C for 45 s, and a final extension at 72 °C for 8 min. Samples were kept at 4 °C, and agarose gel electrophoresis was used to identify representative PCR analysis results. The fragment patterns were then visualized under U.V. using a gel documentation system.

### 2.5. DNA Sequencing and Polymorphism Detection

Primer dimers, nonspecific bands, and other impurities were removed prior to DNA sequencing. Following the manufacturer’s instructions, a PCR purification kit (Jena Bioscience # pp–201 s/ Jena, TH, Germany) was used to purify PCR products of the expected size (target bands), as described by Boom et al. [35]. Nanodrop (Uv-Vis spectrophotometer Q5000/USA) was used to quantify the PCR product in order to ensure sufficient concentrations and purity of the PCR products and to produce high-quality products [36]. In all studied lambs of Barki and Aboudeleik breeds, PCR products of 120 (60 in each breed) with the target band were sent for forward and reverse DNA sequencing in order to find SNPs of the investigated genes.

PCR was carried out for amplification of *CAST* (354-bp), *LEP* (432-bp), *MYLK4* (416-bp), *MEF2B* (424-bp), *STAT5A* (644-bp), *TRPV1* (496-bp), *HSP90AB1* (526-bp), *HSPB6* (630-bp), *HSF1* (480-bp), *ST1P1* (540-bp), and *ATP1A1* (299-bp) genes. Using an ABI 3730XL DNA sequencer (Applied Biosystems, Waltham, MA, USA), amplified fragments were sequenced using the enzymatic chain terminator method established by Sanger et al. [37]. Software such as Chromas 1.45 and BLAST 2.0 was used to evaluate DNA sequencing data [38]. SNPs were identified as differences between the PCR products of the investigated genes and GenBank reference sequences. The MEGA4 program was used to identify differences in the amino acid sequence for the amplified PCR fragments of examined genes between the enrolled lambs based on a sequence alignment [39].

### 2.6. Physical Parameters

Temperature-humidity index (THI) was calculated with the formula described by Amundson et al. [40].
THI = 0.8 × AT °C + [(RH% ÷ 100) × (AT °C − 14.4)] + 46.6(1)
where AT = Ambient temperature, RH = Relative humidity. Throughout the experiment, AT, RH, and THI were calculated at 6:00 am and 2:00 pm (time of physiological responses).

Rectal temperature was assessed using a digital thermometer. Four shaved areas had their skin temperatures checked using a non-contact infrared thermometer (right and left shoulder with right and left hips). The number of abdominal movements per minute was counted while observing the flank movement to determine the respiratory rate.

### 2.7. Hematological Parameters

Blood was collected from the jugular vein into tubes containing EDTA as anticoagulant and placed in ice for estimation of RBCs using an automated instrument for red blood cell (RBCs) and mean corpuscular volume (MCV) counts (Vet-Scan HM2™ Hematology System, Abaxis, Union City, CA, USA) as described by Okoruwa and Ikhimioya [41].

### 2.8. Estimation of Pro-Inflammatory Cytokines

Clear Sera were Collected then Aliquoted and Kept Frozen at −20 °C for Interleukin 1 Beta (IL-1β) and interleukin 6 (IL-6) analysis. The following commercial kits were used according to the standard protocols of the suppliers to quantify the serum concentrations of IL-1β and IL-6 by Aldrich-Sigma Company (St. Louis, MO, USA), ELISA kits [42,43,44].

### 2.9. Economic Evaluation

The economic parameters included total variable costs (TVC), total fixed costs (TFC), total costs (TC), total return (TR), net return (NR), difference ratio in net profit, and economic efficiency. The economic parameters were indicated individually for each breed.

#### 2.9.1. Total Variable Costs (TVC)

Labor, feed, animal care, veterinarian management, production-related expenses, and other costs were included in the total variable costs (TVC) [45].

#### 2.9.2. Total Fixed Costs (TFC)

Depreciation of land, buildings, and equipment was included in the total fixed costs (TFC). The depreciation rate for buildings was computed using a 25-year time frame, while the depreciation rate for equipment was determined using a 5-year time frame [46].
Depreciation rate = value of asset/age of asset (year)

#### 2.9.3. Total Costs (TC)

The sum of the total fixed costs and total variable costs was included in the total costs (TC) [47].
TC = TVC + TFC.

#### 2.9.4. Total Return (TR)

The equation provided by Fidan, 2010 [48] was used to calculate the total return. Additionally, all prices were established in accordance with the going rate during the time of the study.
TR = market price of kg sheep meat × total body weight gain

#### 2.9.5. Net Return (NR)

The net return was calculated by the following equation [49]:NR = total return − total costs.

The difference between the net returns for the two sheep breeds was used to calculate the reduction percentage of net profit. The percentage is calculated by dividing the latter difference by the breed with the highest net return [50].

#### 2.9.6. Economic Efficiency

The ratio between the return from the sale of final body weight together with weight gain and the overall cost of feed consumption was used to calculate the economic efficiency for the sheep breeds. Based on the market pricing of feed ingredients and sheep meat, the cost of each kg of diets and the profits from sales of weight growth were determined. The following equation [51] was used to calculate economic efficiency:(2)$$\mathrm{Economic}\;\mathrm{efficiency}\;\%=\frac{\mathrm{net}\;\mathrm{return}\;\left(\mathrm{EGP}\right)}{\mathrm{total}\;\mathrm{feed}\;\cos t\;\left(\mathrm{EGP}\right)}\times 100$$
where: net return = return of weight (EGP)−total cost (EGP); return of weight (EGP) = total weight × price of kg live BWt (EGP); total feed cost (EGP) = total feed intake (kg/head) × price of kg feed (EGP)

### 2.10. Statistical Analysis

**H****_0_.** *Genetic polymorphisms and economic parameters are not associated with growth and heat tolerance in Barki and Aboudeleik sheep*.

**H****_1_.** *Genetic polymorphisms and economic parameters are associated with growth and heat tolerance in Barki and Aboudeleik sheep*.

Chi-square analysis was used to determine whether there was a statistically significant difference in the detected SNPs in the Barki and Aboudeleik lambs’ genes.

The statistical analysis was made using SPSSPC ± Version 21 by using the Bonferroni test to compare the effects of different genes on growth performance traits. Using the least squares of the general linear model (GLM) methods of SPSS software, an association of discovered SNPs with growth performance and economic parameters was investigated. The model used was as follows:y_ijk_ = µ + G_i_ + B^j^ + e_ijk_(3)
where y_ijk_ is the value of the studied trait, µ is the overall mean of the population, G_i_ is the fixed effect of gene SNP, B^j^ is the fixed effect of breed and e_ijk_ is the random error effect.

Similarly, to explore the effect of nucleotide sequence variants in heat tolerance genes on physico-chemical parameters, the following model is used
y_ijk_ = µ + G_i_ + B^j^ + THI + e_ijk_(4)
where y_ijk_ is the value of the studied trait, µ is the overall mean of the population, G_i_ is the fixed effect of gene SNP, B^j^ is the fixed effect of breed, THI is the fixed effect of temperature humidity index and e_ijk_ is the random error effect.

## 3. Results

### 3.1. Growth Performance of Barki and Aboudeleik Lambs

Table 3 shows the growth performance criteria for Barki and Aboudeleik lambs from the third to the sixth month of age. The initial BW, final BW, final BWG, daily weight gain, and percent increase in BW/month of Barki lambs were all significantly higher than those of Aboudeleik.

### 3.2. PCR-DNA Sequencing of Growth and Heat Stress Genes

All the animals included in the study had high -quality DNA samples available for PCR-DNA sequencing analysis. Nucleotide sequence differences in the form of SNPs were found between Barki and Aboudeleik lambs using PCR-DNA sequencing of the *CAST* (354-bp), *LEP* (432-bp), *MYLK4* (416-bp), *MEF2B* (424-bp), *STAT5A* (644-bp), *TRPV1* (496-bp), *HSP90AB1* (526-bp), *HSPB6* (630-bp), *HSF1* (480-bp), *ST1P1*(540-bp), and *ATP1A1* (299-bp).

Concerning the growth -related genes; DNA sequencing of *CAST* gene (354-bp) revealed one SNP (C196T). Nucleotide sequence variation of *LEP* (432-bp) elicited three SNPs (A69G, C89A, and G305A). Two SNPs (C44A, A94G, C172T, and C249T) were discovered in the sequencing of *MYLK4* (416-bp). Regarding DNA sequencing of *MEF2B* (424-bp), C55T, G157A, and C297G SNPs were denoted. A134G SNP was detected for *STAT5A* (644-bp) gene. DNA sequencing of *TRPV1* (496-bp) elaborated two SNPs (A235G and C434T). All detected SNPs were characteristic of a number of Barki and Aboudeleik lambs.

Regarding the heat tolerance genes, *HSP90AB1* (526-bp) gene sequencing elicited A118G, C232T and A378G SNPs. C155T SNP was found by sequencing the *HSPB6* (630-bp) gene. DNA sequencing of *HSF1* (480-bp) indicated G170A, G283A, and C410T SNPs. For *ST1P1* (540-bp) gene, four SNPs (A177G, C336T, A457G, and C491T) were illustrated. Nucleotide sequence variants of the *ATP1A1* (299-bp) gene displayed A47G and C143T SNPs. All discovered SNPs were unique for a number of Barki and Aboudeleik lambs.

All detected SNPs and their corresponding amino acids were validated by nucleotide and amino acid sequence differences between the studied genes in all the research animals and reference sequences found in GenBank (Appendix A). 

### 3.3. Association of Growth-Related Genes with Growth Performance in Barki and Aboudeleik Lambs

Lambs from each breed were divided into distinct groups according to the presence and harboring of discovered SNPs in genes (group SNPs) related to growth (Table 4). SNPs in the *CAST* gene were used to categorize Barki lambs into GB1*CAST* and GB2*CAST*. Lambs from Aboudeleik were depicted by GA*CAST*. The Barki lambs were divided into two groups, GB1LEP and GB2LEP, based on the discovered SNPs in the *leptin* gene. The discovered SNPs of the *leptin* gene for Aboudeleik lambs divided them into three groups: GA1*LEP*, GA2*LEP*, and GA3*LEP*.

In a similar vein, the discovered SNPs of the *MYLK4* gene divided Barki lambs into three groups: GB1*MYLK4*, GB2*MYLK4*, and GB3*MYLK4*. Two groups, GA1*MYLK4* and GA2*MYLK4*, both represented Aboudeleik lambs. In terms of the *MEF2B* gene, the detected SNPs divided the Barki lambs into GB1*MEF2B* and GB2*MEF2B* and Aboudeleik lambs into GA1*MEF2B* and GA2*MEF2B*. Barki lambs belonged to the GB1*STAT5A* and GB2*STAT5*A *STAT5A* gene groups, whereas Aboudeleik lambs belonged to the GA*STAT5A* group. While Aboudeleik lambs were placed in the GA*TRPV1* group, *TRPV1* divided Barki lambs into five groups: GB1*TRPV1*, GB2*TRPV1*, GB3*TRPV1*, and GB5*TRPV1*. The growth features of the GB1*CAST*, GB2*LEP*, GB1*MYLK4*, GB1*MEF2B*, GB2*STAT5A*, and GB2*TRPV1* groups of Barki lambs were significantly higher than those of the other category groups, according to least squares means of SNPs discriminated groups (*p* ≤ 0.01). Between Barki and Aboudeleik lambs, there was a substantial variation in the frequency of genes related to growth, according to a chi-square analysis of discovered SNPs (Table 4).

### 3.4. Association of Heat Tolerance Genes with Physico-Chemical Parameters in Barki and Aboudeleik Lambs

Barki and Aboudeleik lambs were distinguished into different groups according to nucleotide sequence variants in heat tolerance genes (Table 5). Genetic polymorphisms in *the HSP90AB1* gene categorized Barki lambs into GB1*HSP90AB1* and GB2*HSP90AB1* groups, while Aboudeleik lambs were placed in the GA*HSP90AB1* group. *HSPB6* gene polymorphisms allocated Barki lambs into the GB*HSPB6* group, whereas Aboudeleik was represented by GA1*HSPB6* and GA2*HSPB6* groups. *HSF1* nucleotide sequence variants divided Barki lambs into GB1*HSF1* and GB2*HSF1* groups; meanwhile, Aboudeleik lambs experienced GA1*HSF1*, GA2*HSF1* and GA3*HSF1*.

In the same line, Barki lambs were assigned into GB1*ST1P1* and GB2*ST1P1* groups; however, Aboudeleik lambs exhibited by GA1*ST1P1*, GA2*ST1P1*, GA3*ST1P1* and GA4*ST1P1* according to SNPs detected in *ST1P1* gene. *ATP1A1* gene variants distinguished lambs of each breed into two groups, where Barki and Aboudeleik lambs were illustrated by GA1*ATP1A1*, GA2*ATP1A1*, GB1*ATP1A1* and GB2*ATP1A1* groups.

According to least squares means of SNPs allocated groups (*p* ≤ 0.01), there was a significant association between heat tolerance genes and physico-chemical characters in Barki and Aboudeleik lambs. GB2*HSP90AB1*, GB*HSPB6*, GB2*HSF1*, GB1*ST1P1*, and GB2*ATP1A1* groups of Barki lambs displayed higher skin temperature, respiratory rate, red blood cells (RBCs) count, mean corpuscular volume (MCV), IL-1β and IL-6 than other differentiated groups. However, rectal temperature elicited a different pattern. Between Barki and Aboudeleik lambs, there was a substantial variation in the frequency of genes related to heat tolerance, according to a chi-square analysis of discovered SNPs (Table 5).

### 3.5. Economic Evaluation of Barki and Aboudeleik Lambs

Table 6 lists the economic evaluation criteria for lambs of both Barki and Aboudeleik breeds. The predicted parameters varied significantly between breeds. Compared to Aboudeleik, Barki lambs evoked higher TVC, TC, TR, and NR values. The Barki sheep provided the best economic efficiency value (76%), whereas the Aboudeleik sheep provided a value of 36% in terms of economic efficiency.

Table 7 shows the association between Barki and Aboudeleik lambs’ economic indicators and nucleotide sequence variations in genes relevant to growth. According to least squares means of SNPs discriminated groups, the net returns and economic efficiency percent of the GB1*CAST*, GB2*LEP*, GB1*MYLK4*, GB1*MEF2B*, and GB2*TRPV1* SNPs identified groups of Barki lambs were significantly higher than those of other assigned groups.

## 4. Discussion

In numerous nations with a sophisticated sheep-raising industries, DNA marker-based technologies are widely used in national breeding programs and have a considerable impact on improving growth traits [52]. The development of methods for better utilizing the gene pools of current breeds of sheep, reducing feed costs, putting genetic control mechanisms in place, finding more reserves, and improving the sector’s economic performance are the key issues facing sheep breeding globally [2]. Given the importance of molecular-genetic studies, it is imperative to assess the potential genes responsible for sheep’s productive traits [53].

Our findings demonstrated a substantial difference in final BW and BWG between the lambs of the two breeds, with Barki lambs having greater final BW and BWG than Aboudeleik. A larger percentage of BWG every month in Barki lambs, may be explained by their higher ultimate BW and BWG. Our findings correspond with those of Elnageeb and [28] and [54], who described Barki sheep as having high growth and nutritional performance. The significant differences in final BW and BWG are mostly caused by the different daily feed intakes between the two breeds [55].

In this study, a PCR-DNA sequencing technique was used to characterize the growth and heat tolerance genes in lambs from Barki and Aboudeleik at the molecular level. The findings showed that there were SNPs among the lambs of the two breeds. When compared to the corresponding GenBank reference sequence, it is interesting to note that the polymorphisms found and published here reveal new insights about the studied genes. Our findings showed that the growth genes with detected SNPs separated lambs from each breed into distinct groups, where the GB1*CAST*, GB2*LEP*, GB1*MYLK4*, GB1*MEF2B*, GB2*STAT5A*, and GB2*TRPV1* groups of Barki lambs exhibited significantly higher growth performance traits than other groups. As a result, the genes *CAST*, *LEP*, *MYLK4*, *MEF2B*, *STAT5A*, and *TRPV1* might be utilized as proxy biomarkers for sheep growth performance. Previous research expanded on the relationship between sheep growth features and polymorphisms in the *CAST*, *LEP*, *MEF2B*, and *STAT5A* genes [56,57,58]. Our study is notable since it is the first to describe genetic variants in the *MYLK4* and *TRPV1* genes and their relationship to growth features in sheep.

Quantitative growth features are regulated by a number of genes [59]. For instance, calpastatin (*CAST*) has a remarkable impact on growth parameters and carcass features [60]. The structure and location of the *CAST* gene in sheep have been determined; its locus is on chromosome 5, has 29 exons, and measures 89,553 bp [61]. It has been established that sheep *CAST* gene variation affects both growth traits and meat qualities [62]. A prospective candidate gene for QTL studies is leptin, a protein closely linked to animal growth and metabolism [63]. Additionally, it controls the amount of food consumed as well as how the body uses its energy [64]. The polymorphisms of the ovine *leptin* gene and their correlation with growth and carcass features have been documented [63,64].

The *MYLK4* gene, a member of the MYLK family, may be essential for the growth of muscles [65]. The *MYLK4* gene has been shown to be a promising candidate gene for enhancing the growth characteristics of cattle [65]. In a similar vein, Shi et al. [66] identified a link between the goat *MYLK4* gene and growth features. Myocyte enhancer factor-2 (MEF2B), which is a key player in the control of muscular growth and development, is another candidate gene for growth [67]. The relationship between sheep growth features and *MEF2B* gene variations was described [57,68]. The major mediator of growth hormone activity on target genes is known as signal transducer and activator of transcription 5 (*STAT5*). It is essential as an intracellular modulator of prolactin signaling and can cause milk protein gene transcription in response to prolactin [69]. Cattle are the main subject of studies on *STAT5A* polymorphisms and growth traits, and milk production traits [70]. In sheep, the *STAT5A* gene was evaluated as a surrogate marker for growth [58].

A variety of environmental data are normally transmitted to the inside of the cell by the transient receptor potential (TRP) family of channels using calcium entry [71]. Specific TRP channels react to stimuli like high and low temperatures, anisotonicity, various tastants, and a range of other stimuli [72]. TRP channels are divided into six main families based on sequence homology [73]; the TRPV family is named after its founding member, the vanilloid receptor, or TRPV1. The vanilloid or capsaicin receptor, also known as TRPV1, was originally molecularly discovered and cloned as a nonselective cation channel [74,75]. TRPV1 mediates the ion fluxes that occur when neurons and mucosae are exposed to various unpleasant stimuli [71]. Our study is the first to determine the effects of *TRPV1* gene polymorphisms on growth traits in sheep. Our study was based on earlier work linking the variations and haplotypes of the *TRPV1* gene to features related to cattle growth [76].

There is little information about the molecular mechanism of heat resistance in ruminants. However, in the summer, high temperatures can result in mild heat stress, which causes oxidative stress [77], and in mid-lactating cows, minor adverse effects of heat stress on a number of plasma indicators of oxidative state were identified. When compared to Aboudeleik lambs, nucleotide variation was found in the *HSP90AB1* (526-bp), *HSPB6* (630-bp), *HSF1* (480-bp), *ST1P1* (540-bp), and *ATP1A1* (299-bp) genes by PCR-DNA sequencing. Our findings indicated a significant difference in the distribution of heat tolerance genes in lambs of the two breeds. Rare reports exist about the molecular characterization of the ovine *HSP90AB1* gene [32]. To our knowledge, no studies have previously explored the relationships between the sheep *HSPB6*, *HSF1*, *ST1P1*, and *ATP1A1* genes.

Heat stress is characterized by the transcription of a group of proteins known as heat shock proteins (HSP) [78]. According to their molecular weight, ’the isoforms of these proteins are divided into families, including HSP27, HSP60, HSP70, HSP90, and HSP110/104 [78]. As a result of heat or stress induction, 90-kilodalton heat shock proteins (HSP90) are constitutively produced and serve as significant molecular chaperones [79]. Gene duplication produces the constitutive and inducible versions of the two main cytoplasmatic HSP90 isoforms [80]. Numerous studies have been done on the role of HSP90 isoforms in a number of cellular functions, such as signal transduction, protein folding, protein degradation, cell survival, and morphological evolution [79]. In sheep [32] and cattle [80], *HSP90AB1* has been linked to features of adaptation and heat tolerance. The *HSPB6* gene has also been shown to be related to features of heat -resistant cattle [81].

The *HSF1* gene is highly conserved in eukaryotes and serves as the principal medium of protein toxic stress transcription response [82]. It also plays a crucial role in the non-stress regulation of development and metabolism [83]. When compared to those with lesser activity, the *HSF1* gene is more protective against heat stress [84,85]. As a result, it was assumed that the *HSF1* gene controlled heat shock [86]. In Chinese cattle, Rong et al. [87] found a link between *HSF1* genetic variation and heat tolerance. The best -suited examples of genes whose expression is affected by heat stress are those encoding heat shock proteins (HSP), but it has recently become clear that thermal stress also induces a significant number of genes that are not often thought of as *HSP* genes. It has been discovered that 50 genes whose expression has not previously been associated with *HSPs* change in response to heat stress [79]. As an illustration, HSP70 and HSP90 play coordinated functions in protein folding under the control of the adaptor protein STIP1. Binding both HSP90 and substrate-bound HSP70 is expected to aid in the transfer of proteins from HSP70 to HSP90 [78]. Additionally, STIP1 promotes the ATPase activity of HSP70 while inhibiting the ATPase activity of HSP90, indicating that it controls both these chaperones’ conformations and ATPase cycles [86].

Further research revealed that heat stress affected the plasma Na+ and K+ levels in ruminants [88]. We chose the Na+, K± ATPase gene as a candidate for heat shock response because it is particularly sensitive to oxidative stress and is in charge of creating the electrochemical gradient of Na+ and K+ across the plasma membrane, where the ion gradients formed by the enzyme are required for Na± coupled transport. This was done in order to better understand the molecular mechanism of heat shock response [89]. In this study, we first present the discovery of two unique SNPs in the ovine *ATP1A1* gene, which may help to explain the differences in heat tolerance between the two sheep breeds. It is interesting to note that genetic variants in the *ATP1A1* gene have been linked to cattle heat tolerance [89,90].

Our findings revealed that there was a significant association between heat tolerance genes and physico-chemical characters in Barki and Aboudeleik lambs. GB2*HSP90AB1*, GB*HSPB6*, GB2*HSF1*, GB1*ST1P1*, and GB2*ATP1A1* groups of Barki lambs displayed higher skin temperature, respiratory rate, red blood cells (RBCs) count, mean corpuscular volume (MCV), IL-1β and IL-6 than other differentiated groups. However, rectal temperature elicited a different pattern. Our study is the first that reports the association of heat tolerance genes polymorphism with physico-chemical parameters in livestock, particularly sheep. Despite possessing a highly developed thermoregulation mechanism, ruminants do not maintain strict homeothermy under stress [91]. The integration of numerous organs and systems, including the immune, endocrine, cardio-respiratory, and behavioral systems, results in a variety of physiological and behavioral responses to thermal stress that vary in intensity and duration depending on the genetic make-up of the animal and environmental factors [92].

Body temperature is regulated by cytokines, which are defined as regulatory proteins of polypeptides produced by immune cells in response to tissue injury, infection, stress, or inflammation [92]. Rectal temperature is the most frequent measure of body temperature and is thought to represent core body temperature as well as physiological homeothermy characteristics, leading to an adaptive slowing of the metabolic rate and decreased appetite [93,94]. When subjected to high temperatures, animals maintain the thermal equilibrium congener’s dispersion of surplus heat from their bodies. Numerous biological processes are followed by this, with an increase in breathing rate or panting being the most evident response. When the body is unable to maintain thermal balance in these conditions, the animal’s body temperature increases [10]. Additionally, one of the physiological reactions of animals exposed to heat stress during the summer was a considerable increase in skin temperature (ST) [93]. Due to the vasodilatation of the skin’s capillary bed and the ensuing increase in blood flow to the skin’s surface, exposed animals’ ST increases during the summer [95,96].

To enhance the body’s response to heat stress, blood flow has to be increased. As a compensatory measure to prevent shock, this increase in blood flow was accomplished via an increase in cardiac function and blood vessels [97]. The animals’ response to heat stress was an increase in heart rate, which caused them to pump more blood from the heart to the skin’s surface, increasing the likelihood that sensible or active (heat loss through conduction, radiation, and convention) and insensible or passive (heat was used to vaporize water) heat loss would occur [10]. In order to quantify the harmful effects of the environmental temperature, respiratory rate (RR) was used as a measure of heat stress [98]. The rise in RR during the summertime suggested heat stress, and panting was one of the animals’ ways of releasing extra heat through evaporation [10]. High environmental temperature increases the oxygen consumption of animals by increasing the respiration rate, which results in an increased number of erythrocytes as a compensatory mechanism [99].

The cytokine system, and particularly IL-6, mediates the short-term transitory increase in heat shock proteins following exposure [100]. This is helpful in a number of adaptive benefits of heat stress on extensive exposure-induced changes in cytokines and total antioxidant capacity (TAC) as well as inflammation parameters that heat may be successfully attained in a short amount of time [100]. This inhibits the formation of free radicals and lessens their harmful effects, particularly on the cell membrane [99]. The hypothalamic-pituitary-thyroid axis (HPT) and hypothalamic-pituitary-adrenal axis (HPA) are known to play crucial roles in how mammals adapt to changes in their environment, and IL-6 acts as the primary regulator of the acute-phase protein response [101]. IL-1β is a powerful cytokine that promotes inflammation and functions as an endogenous pyrogen. After brain damage or peripheral immune activation, it has been found in the CNS [101]. Thyroxine (T4) and triiodothyronine (T3), which are recognized to be crucial in helping mammals adapt to changes in their environment, are decreased in plasma levels as a result of IL-1 acting to suppress thyroid function [99]. Our findings and the aforementioned information lead us to hypothesize that the Barki breed of sheep is more heat tolerant than Aboudeleik.

There is a significant variation in TVC and TC values amongst lambs of the two sheep breeds when it comes to the parameters used for economic evaluation. The Barki lambs demonstrated the greatest values of TVC, TC, TR, and NR. The substantial discrepancy between TVC and TC is attributable to the different feed costs, which make up the majority of manufacturing costs. Feed prices also have an impact on overall costs as well as total variable costs [29].

The total and net returns revealed highly significant variations between the lambs of the two sheep breeds in terms of the results of returns. Because TR is dependent on the final body weight and body weight gain of the sheep breed, there are discrepancies between the TR and NR values. Because Barki lambs have higher final BW and BWG values than Aboudeleik lambs and because both breeds’ market meat prices are equal, Barki has higher returns than Aboudeleik. Additionally, TR includes the TC in its calculation. Furthermore, NR is the remaining net revenue or profit after deducting the TC, which likewise varies significantly amongst sheep breeds [102]. Regarding the disparity between net profit and economic efficiency, Barki sheep provided the best economic efficiency value (76%), whereas Aboudeilik sheep provided 36%. The difference in net profit between Barki and Aboudeilik sheep was the same, showing that Barki had a net profit that was nearly 58% higher. This is explained by the fact that the Barki breed has the highest net return, which affects the calculation of economic efficiency [103]. Our study used a novel approach to investigate the relationship between nucleotide sequence variations in genes related to growth and economic parameters in Barki and Aboudeleik lambs. Where the net returns and economic efficiency % of the GB1*CAST*, GB2*LEP*, GB1*MYLK*4, GB1*MEF2B*, and GB2*TRPV1* SNPs identified groups of Barki lambs were significantly greater than other assigned groups.

In species that have evolved in harsh settings, tolerance to high heat and humidity is an essential functional feature [104]. It is regulated by molecular pathways that show the intricate relationship between genes and the environment [105]. It becomes crucial to consider the relationship between body size and heat stress tolerance [106]. As a response to the tropical temperature that prevailed in their ancestral habitat, smaller size and resistance to heat stress evolved together [107]. These genotypes are therefore regarded as a significant genetic resource for selection for climate resistance [108]. Stressed animals have up-regulated stress response and protein repair genes while down -regulating biosynthesis, metabolism, and body conformation genes [108,109].

## 5. Conclusions

Further studies should be done on a large number and different breeds of sheep. Based on our results, we can speculate that the identified SNPs in growth and heat tolerance genes had a strong correlation with growth performance and heat tolerance features. In order to characterize the two breeds, the variance indicated could be utilized as a proxy marker, allowing for marker-assisted selection for growth and heat tolerance traits in sheep. We can assume that Barki sheep are superior to Aboudeleik sheep economically.

## Figures and Tables

**Table 1 animals-13-00353-t001:** Composition of the concentrate feed mixture (CFM) fed to growing Barki and Aboudeleik lambs.

Ingredients	Quantity (kg)
Corn	400
Wheat bran	300
Soya beans	250
Sodium chloride	10
Calcium carbonate	20
Premix	1
Netro-Nill	0.5
Fylax	0.5

**Table 2 animals-13-00353-t002:** Forward and reverse primer sequence, length of PCR product and annealing temperature for growth and heat tolerance genes used in PCR-DNA sequencing.

Gene	Forward	Reverse	Accession Number	Annealing Temperature (°C)	Length of PCR Product (bp)	Reference
*CAST*	5′-GCTGGGTCTTCATTGCTGTGT-3′	5′-TGACTGGCTGATGAAGGAATTG-3′	LC717976.1	60	354	Current study
*LEP*	5′-ACTGGTTTGGACTTCATCCCTGG-3′	5′-GCTGGCCTGCATAAAGGATGCC-3′	AH014693.2	62	432	
*MYLK4*	5′-ATGGGAGCTGTTTCTCCCTCTTG-3′	5′-TGCCTGGCCGTCACGATGCGGT-3′	XM_027958720.2	62	416	
*MEF2B*	5′-GGTAACATTCACCAAGCGGAAGT-3′	5′-CTCCCCAAGTCCAGTGGGTTCAC-3′	XM_042249577.1	64	424	
*STAT5A*	5′-TGGGATGCCATCGACCTGGACAA-3′	5′-CTGGGACATGGCATCAACCAGGA-3′	NM_001009402.2	62	644	
*TRPV1*	5′-TTAAGACTGAGGGAAGTGGGTGC-3′	5′-CCTGTGAAGTGCCTCAGGTGAAC-3′	XR_006061143.1	60	496	
*HSP90AB1*	5′-GTGCTTCGCCTTATATAAAGTGA-3′	5′-CTGCCCAATCATGGAGATGTCT-3′	XM_004018854.5	60	526	
*HSPB6*	5′-ACTGCGGAGCCTAGGGCGACGGC-3′	5′-TAGTCTTGGCGCGCGCTGGGGAC-3′	XM_027977443.2	64	630	
*HSF1*	5′-GGCCTCTCTTTGCGGTTGCTAC-3′	5′-TCTCATGCTTCATGGCCAGCAG-3′	XM_027973370.2	60	480	
*ST1P1*	5′-GAATAACGGAGGCAGAGCCCCTT-3′	5′-CAATTGGTCGGTGGTTGCAGAC-3′	XM_042238340.1	62	540	
*ATP1A1*	5′-CCTGCGTGGTGCATGGAAGTGATC-3′	5′-CTCCAGGCGTAGAGGATGGTGAC-3′	KP325220.11	60	299	

*CAST* = calpastatin; *LEP* = leptin; *MYLK4* = myosin light chain kinase family member 4; *MEF2B* = myocyte enhancer factor 2B; *STAT5A* = signal transducer and activator of transcription 5A; *TRPV1* = transient receptor potential cation channel subfamily V member 1; *HSP90AB1* = heat shock protein 90 alpha family class B member 1; *HSPB6* = heat shock protein family B (small) member 6; *HSF1* = heat shock transcription factor 1; *ST1P1* = stress -induced phosphoprotein 1 and *ATP1A1* = ATPase Na+ / K+ transporting subunit Alpha 1.

**Table 3 animals-13-00353-t003:** Growth performance criteria of Barki and Aboudeleik lambs from the third to the sixth month of age.

Growth Parameters	Sheep Breed	
	Barki	Aboudeleik
Initial BW (kg)	27.65 ± 0.36	26.92 ± 0.24
Final BW (kg)	41.06 ^a^ ± 0.51	36.57 ^b^ ± 0.44
Final BWG (kg)	13.41 ^a^ ± 0.18	9.65 ^b^ ± 0.13
% of increase in BW/month	16.2%	11.9%
Daily weight gain (DWG) (kg/day)	0.149 ^a^ ± 0.003	0.107 ^b^ ± 0.007

BW = body weight; BWG = body weight gain; DWG = daily weight gain; and FCR = feed conversion ratio. Means of different levels within the same row having different superscripts are significantly different (*p* < 0.05).

**Table 4 animals-13-00353-t004:** Distribution of SNPs, type of mutation in growth-related genes and their association with growth performance in Barki and Aboudeleik lambs from the third to the sixth month of age.

Gene	SNP	Amino Acid Number and Type	Breed	Number of Lambs Harboring SNP	Group SNP	Chi Square	Initial BW (kg)	Final BW (kg)	Final BWG (Kg)	% of Increase in BW/month	Daily Weight Gain (kg/day)
*CAST*	C196T	66 R to W Non-synonymous	Barki	37	GB1*CAST*	77.76 ** (*p* = 0.007)	28.10 ^a^ ± 0.31	42.01 ^a^ ± 0.39	13.91 ^a^ ± 0.47	16.5	0.154 ^a^ ± 0.002
	-	-		23	GB2*CAST*	48.34 ** (*p* = 0.006)	27.20 ^b^ ± 0.23	40.11 ^b^ ± 0.41	12.91 ^b^ ± 0.20	15.8	0.143 ^a^ ± 0.004
	-	-	Aboudeleik	60	GA*CAST*	126.09 ** (*p* = 0.008)	26.92 ^b^ ± 0.24	36.57 ^c^ ± 0.44	9.65 ^c^ ± 0.13	11.9	0.107 ^b^ ± 0.007
*LEP*	A69G	23 A Synonymous	Barki	28	GB1*LEP*	58.84 ** (*p* = 0.006)	27.33 ^b^ ± 0.28	39.96 ^b^ ± 0.35	12.63 ^b^ ± 0.19	15.4	0.140 ^a^ ± 0.005
	-	-		32	GB2*LEP*	67.25 ** (*p* = 0.006)	27.97 ^a^ ± 0.18	42.16 ^a^ ± 0.22	14.19 ^a^ ± 0.27	16.9	0.157 ^a^ ± 0.004
	C89A, G305A	30 A to D Non-synonymous 102 R to Q Non-synonymous	Aboudeleik	41	GA1*LEP*	86.17 ** (*p* = 0.005)	27.58 ^b^ ± 0.11	37.62 ^c^ ± 0.29	10.04 ^c^ ± 0.29	12.1	0.111 ^b^ ± 0.003
	G186A	62 L Synonymous		10	GA2*LEP*	21.01 ** (*p* = 0.006)	26.46 ^c^ ± 0.22	35.82 ^d^ ± 0.33	9.36 ^d^ ± 0.25	11.8	0.104 ^b^ ± 0.005
	-	-		9	GA3*LEP*	18.91 ** (*p* = 0.006)	26.74 ^c^ ± 0.19	36.27 ^d^ ± 0.37	9.53 ^d^ ± 0.34	11.9	0.105 ^b^ ± 0.005
*MYLK4*	A94G	32 T to A Non-synonymous	Barki	35	GB1*MYLK4*	73.56 ** (*p* = 0.006)	28.24 ^a^ ± 0.24	41.95 ^a^ ± 0.17	13.71 ^a^ ± 0.20	16.2	0.152 ^a^ ± 0.007
	C172T	58 R to C Non-synonymous		8	GB2*MYLK4*	16.81 ** (*p* = 0.005)	27.49 ^b^ ± 0.40	40.81 ^b^ ± 0.32	13.32 ^b^ ± 0.37	16.2	0.148 ^a^ ± 0.010
	-	-		17	GB3*MYLK4*	35.72 ** (*p* = 0.009)	27.24 ^b^ ± 0.29	40.43 ^b^ ± 0.21	13.19 ^b^ ± 0.31	16.1	0.147 ^a^ ± 0.009
	C44A C249T	15 A to D Non-synonymous 83 S Synonymous	Aboudeleik	44	GA1*MYLK4*	92.47 ** (*p* = 0.005)	27.30 ^b^ ± 0.26	37.01 ^c^ ± 0.25	9.71 ^c^ ± 0.28	11.8	0.108 ^b^ ± 0.011
	-			16	GA2*MYLK4*	33.62 ** (*p* = 0.009)	26.54 ^c^ ± 0.35	36.13 ^d^ ± 0.31	9.59 ^c^ ± 0.32	12	0.106 ^b^ ± 0.006
*MEF2B*	C55T	19 R to C Non-synonymous	Barki	37	GB1*MEF2B*	77.76 ** (*p* = 0.006)	27.99 ^a^ ± 0.29	41.72 ^a^ ± 0.24	13.73 ^a^ ± 0.27	16.4	0.153 ^a^ ± 0.012
	-	-		23	GB2*MEF2B*	48.34 ** (*p* = 0.009)	27.31 ^b^ ± 0.36	40.40 ^b^ ± 0.34	13.09 ^b^ ± 0.36	16	0.145 ^a^ ± 0.008
	G157A C297G	53 G to R Non-synonymous 99G Synonymous	Aboudeleik	38	GA1*MEF2B*	79.86 ** (*p* = 0.004)	27.28 ^b^ ± 0.32	37.43 ^c^ ± 0.17	10.15 ^c^ ± 0.20	12.4	0.113 ^b^ ± 0.010
	-	-		22	GA2*MEF2B*	46.24 ** (*p* = 0.006)	26.56 ^c^ ± 0.23	35.71 ^d^ ± 0.28	9.15 ^d^ ± 0.19	11.5	0.101 ^b^ ± 0.005
*STAT5A*	A134G	45 Q to R Non-synonymous	Barki	18	GB1*STAT5A*	37.83 ** (*p* = 0.008)	27.25 ^b^ ± 0.26	40.38 ^b^ ± 0.43	13.13 ^b^ ± 0.30	16.1	0.146 ^b^ ± 0.002
	-	-		42	GB2*STAT5A*	88.27 ** (*p* = 0.004)	28.05 ^a^ ± 0.37	41.74 ^a^ ± 0.39	13.69 ^a^ ± 0.27	16.3	0.152 ^a^ ± 0.009
	-	-	Aboudeleik	60	GA*STAT5A*	126.09 ** (*p* = 0.002)	26.92 ^b^ ± 0.24	36.57 ^c^ ± 0.44	9.65 ^c^ ± 0.13	11.9	0.107 ^c^ ± 0.007
*TRPV1*	A235G	79 K to E Non-synonymous 145 T to M Non-synonymous	Barki	17	GB1*TRPV1*	35.72 ** (*p* = 0.006)	28.00 ^b^ ± 0.34	41.47 ^b^ ± 0.38	13.47 ^a^ ± 0.23	16	0.150 ^a^ ± 0.010
	C434T			20	GB2*TRPV1*	42.03 ** (*p* = 0.005)	28.35 ^a^ ± 0.29	41.88 ^a^ ± 0.34	13.53 ^a^ ± 0.28	15.9	0.150 ^a^ ± 0.006
	A235G, C434T			9	GB3*TRPV1*	18.91 ** (*p* = 0.009)	27.30 ^c^ ± 0.25	40.65 ^c^ ± 0.37	13.35 ^a^ ± 0.31	16.3	0.148 ^a^ ± 0.003
	-			14	GB5*TRPV1*	29.42 ** (*p* = 0.009)	26.95 ^d^ ± 0.18	40.24 ^d^ ± 0.35	13.29 ^a^ ± 0.16	16.4	0.148 ^a^ ± 0.012
	-	-	Aboudeleik	60	GA*TRPV1*	126.09 ** (*p* = 0.009)	26.92 ^d^ ± 0.24	36.57 ^e^ ± 0.44	9.65 ^b^ ± 0.13	11.9	0.107 ^b^ ± 0.007

*CAST*= calpastatin; *LEP* = leptin; *MYLK4* = myosin light chain kinase family member 4; *MEF2B* = myocyte enhancer factor 2B; *STAT5A* = signal transducer and activator of transcription 5A; *TRPV1* = transient receptor potential cation channel subfamily V member 1. A = Alanine; C = Cisteine; D = Aspartic acid; E = Glutamic Acid; G = Glycine; K = lysine; L = Leucine; M = Methionine; Q = Glutamine; R = Argnine; S = Serine; T = Threonine; and W = Tryptophan. Means within the same column of, different litters are significantly different at (*p* < 0.01). ** = Significant at (*p* < 0.01).

**Table 5 animals-13-00353-t005:** Distribution of SNPs, type of mutation in heat tolerance genes and their association with physico-chemical parameters in Barki and Aboudeleik lambs.

Gene	SNPs	Amino Acid Number and Type	Breed	No of Lambs Harboring SNP	Group SNP	Chi Square	Skin Temperature °C	Rectal Temperatue °C	Respiratory Rate Breath/Min	RBCs Count (10^3^/m^3^)	MCV fL	IL-1β Pg/mL	IL-6 Pg/mL
*HSP90AB1*	A118G A478G	40 I to V Non-synonymous 160 I to V Non-synonymous	Barki	26	GB1*HSP90AB1*	54.64 ** (*p* = 0.006)	36.2 ± 0.4 ^b^	38.4 ± 0.4 ^b^	47.5 ± 3.45 ^a^	1750 ± 45.52 ^a^	58.90^a^ ± 1.74 ^a^	85.62 ± 0.54 ^b^	90.41 ± 0.74 ^b^
	C232T	34		34	GB2*HSP90AB1*	71.46 ** (*p* = 0.0024)	37.5 ± 0.6 ^a^	38.2 ± 0.2 ^b^	47.8 ± 3.89 ^a^	1780 ± 30.60 ^a^	60.42 ± 1.22 ^a^	95.24 ± 0.63 ^a^	112.51 ± 0.62 ^a^
	-	-	Aboudeleik	60	GA*HSP90AB1*	126.09 ** (*p* = 0.005)	36 ± 0.5 ^b^	38.8 ± 0.7 ^a^	46.5 ± 2.85 ^b^	1250 ± 40.30 ^b^	52.34 ± 0.80 ^b^	73.82 ± 0.68 ^c^	81.51 ± 1.47 ^c^
*HSPB6*	-	-	Barki	60	GB*HSPB6*	126.09 ** (*p* = 0.003)	37.8 ± 0.5 ^a^	38.2 ± 0.5 ^b^	48.6 ± 2.60 ^a^	1800 ± 40.54 ^a^	60.80 ± 0.96^a^	108 ± 0.51 ^a^	116.74 ± 0.81 ^a^
	C155T	52 A to V Non-synonymous	Aboudeleik	46	GA1*HSPB6*	96.67 ** (*p* = 0.0015)	36.4 ± 0.5 ^b^	39 ± 0.2 ^a^	46 ± 3.50 ^b^	1320 ± 20.82 ^b^	55.84 ± 0.62 ^b^	92.86 ± 0.41 ^b^	104.35 ± 0.42 ^b^
	-	-		14	GA2*HSPB6*	29.42 ** (*p* = 0.006)	36.7 ± 0.2 ^b^	38.5 ± 0.3 ^b^	46.6 ± 3.50 ^b^	1220 ± 35.78 ^c^	51.63 ± 0.62 ^c^	86.43 ± 0.25 ^c^	91.51 ± 1.62 ^c^
*HSF1*	G283A	95 G to R Non-synonymous	Barki	33	GB1*HSF1*	69.35 ** (*p* = 0.0017)	36.2 ± 0.3 ^a^	38.5 ± 0.4 ^a^	47.5 ± 3.40 ^a^	1480 ± 20.48 ^b^	56.48 ± 0.52 ^b^	78.83 ± 0.68 ^d^	84.42 ± 0.81 ^a^
	-			27	GB2*HSF1*	56.74 ** (*p* = 0.0026)	37.4 ± 0.6^a^	37.8 ± 0.2 ^b^	47.8 ± 3.47 ^a^	1860 ± 47.55^a^	62.45 ± 1.45 ^a^	99.51 ± 1.76 ^a^	90.21 ± 0.73 ^a^
	G170A	57 G to E Non-synonymous	Aboudeleik	16	GA1*HSF1*	33.62 ** (*p* = 0.006)	36.1 ± 0.2 ^b^	38.2 ± 0.6 ^a^	45.80 ± 2.30 ^b^	1370 ± 48.24 ^c^	55.90 ± 0.80 ^a^	82.62 ± 1.51 ^d^	88.12 ± 1.43 ^a^
	G170A C410T	57 G to E Non-synonymous 137 A to V Non-synonymous		31	GA2*HSF1*	65.15 ** (*p* = 0.0015)	36.4 ± 0.4 ^b^	38.6 ± 0.5 ^a^	47.20 ± 3.50 ^a^	1250 ± 50.48 ^d^	52.42 ± 1.26 ^c^	86.55 ± 0.42 ^c^	89.52 ± 1.94 ^b^
	-	-		13	GA3*HSF1*	27.32 ** (*p* = 0.009)	36 ± 0.5 ^b^	38.4 ± 0.2 ^a^	46.3 ± 3.90 ^b^	1310 ± 40.28 ^c^	55.62 ± 0.75 ^b^	87.42 ± 0.61 ^c^	88.52 ± 1.23 ^b^
*ST1P1*	A177G	60 S Synonymous	Barki	47	GB1*ST1P1*	98.77 ** (*p* = 0.0022)	37.5 ± 0.4 ^a^	38.2 ± 0.4 ^b^	48.30 ± 2.50 ^a^	1790 ± 25.88 ^a^	59.41 ± 0.48 ^a^	102.66 ± 0.52 ^a^	114.33 ± 1.47 ^a^
	-	-		13	GB2*ST1P1*	27.32 ** (*p* = 0.008)	36.2 ± 0.3 ^b^	38.6 ± 0.3 ^b^	48 ± 2.30 ^a^	1524 ± 40.83 ^b^	56.82 ± 0.53 ^b^	88.51 ± 1.81 ^c^	95.14 ± 0.52 ^c^
	C336T	112 C Synonymous	Aboudeleik	17	GA1*ST1P1*	35.72 ** (*p* = 0.002)	36.4 ± 0.2 ^b^	38.5 ± 0.2 ^b^	46.70 ± 3.50 ^b^	1210 ± 50.49 ^d^	50.36 ±0.83 ^c^	92.28 ± 0.43 ^b^	102.47 ± 0.57 ^b^
	C336T, C491T	112 C Synonymouf 164 A to V Non-synonymous		20	GA2*ST1P1*	42.03 ** (*p* = 0.006)	36.5 ± 0.3 ^b^	38.8 ± 0.4 ^a^	45.80 ± 2.70 ^c^	1380 ± 30.78 ^c^	55.66 ± 1.62 ^b^	87.92 ± 0.72 ^c^	92.14 ± 0.48 ^d^
	A457, C491T	153 M to L Non-synonymous 164 A to V Non-synonymous		12	GA3*ST1P1*	25.22 ** (*p* = 0.009)	36 ± 0.5 ^b^	38.6 ± 0.4 ^b^	46.20 ± 3.80 ^b^	1330 ± 42.78 ^c^	55.38 ± 0.94 ^b^	72.32 ± 1.24 ^e^	85.62 ± 0.81 ^e^
	C491T	164 A to V Non-synonymous		11	GA4*ST1P1*	23.11 ** (*p* = 0.001)	36.6 ± 0.3 ^b^	38.6 ± 0.5 ^a^	45.70 ± 2.80 ^c^	1520 ± 30.53 ^b^	56.65 ± 0.34 ^b^	80.22 ± 0.48 ^d^	91.41 ± 0.41 ^d^
*ATP1A1*	A47G	16 N to S Non-synonymous	Aboudeleik	21	GA1*ATP1A1*	44.13 ** (*p* = 0.0029)	36 ± 0.4 ^b^	38.4 ± 0.7 ^a^	45.80 ± 2.60 ^c^	1620 ± 26.43 ^b^	57.36 ± 1.62 ^a^	75.91 ± 0.58 ^c^	82.17 ± 0.68 ^c^
	-	-		39	GA2*ATP1A1*	81.96 (*p* = 0.004)	36.2 ± 0.5 ^b^	38.7 ± 0.6 ^a^	46.50 ± 3.40 ^b^	1370 ± 55.43 ^c^	54.71 ± 1.81 ^b^	78.21 ± 1.84 ^d^	83.41 ± 1.62 ^c^
	C143T	48 S to L Non-synonymous	Barki	31	GB1*ATP1A1*	65.15 ** (*p* = 0.0017)	36.4 ± 0.3 ^b^	38.6 ± 0.2 ^a^	46.70 ± 2.90 ^b^	1630 ± 30.76 ^b^	55.84 ± 0.62 ^b^	87.24 ± 0.73 ^b^	93.51 ± 1.72 ^b^
	-	-		29	GB2*ATP1A1*	60.94 ** (*p* = 0.005)	37.4 ± 0.5 ^a^	37.8 ± 0.5 ^b^	48.70 ± 3.50 ^a^	1760 ± 40.58 ^a^	58.72 ± 0.97 ^b^	92.14 ± 1.54 ^a^	109.74 ± 0.45 ^a^

*HSP90AB1* = heat shock protein 90 alpha family class B member 1; *HSPB6* = heat shock protein family B (small) member 6; *HSF1* = heat shock transcription factor 1; *ST1P1* = stress -induced phosphoprotein 1 and *ATP1A1* = ATPase Na +/ K + transporting subunit Alpha 1. A = alanine; C = cisteine; E = glutamic acid; G = glycine; H = histidine; I = isoleucine; L = leucine; M = methionine; N = asparagine; R = argnine; S = serine; V = valine; and Y = tyrosine. Means within the same column of, different litters are significantly different at (*p* < 0.01). ** = Significant at (*p* < 0.01). RBCS = Red blood cells; MCV = mean corpuscular volume; IL-1β = Interleukin-1 beta; IL-6 = Interleukin-6.

**Table 6 animals-13-00353-t006:** Economic evaluation criteria of Barki and Aboudeleik lambs.

Economic Parameters	Sheep Breed	
	Aboudeleik	Barki
TVC	631.67 ^b^ ± 0.45	707.21 ^a^ ± 0.63
TFC	15	15
TC	646.67 ^b^ ± 0.72	722.21 ^a^ ± 0.97
TR	820.25 ^b^ ± 0.85	1139.85 ^a^ ± 0.96
NR	173.58 ^b^ ± 0.37	417.64 ^a^ ± 0.59
Difference % in Net Profit	−0.58	+0.58
Economic efficiency %	0.36	0.76

Means of different levels within the same row having different superscripts are significantly different (*p* < 0.05).

**Table 7 animals-13-00353-t007:** Distribution of SNPs, type of mutation in growth -related genes and their association with economic parameters in Barki and Aboudeleik lambs.

Gene	SNP	Amino Acid Number and Type	Breed	Number of Lambs Harboring SNP	Group SNP	Chi Square	NR (Net Return)	Economic Efficiency %
*CAST*	C196T	66 R to W Non-synonymous	Barki	37	GB1*CAST*	77.76 ** (*p* = 0.007)	423.06 ± 0.46 ^a^	79
	-	-		23	GB2*CAST*	48.34 ** (*p* = 0.006)	412.22 ± 0.63 ^b^	73
	-	-	Aboudeleik	60	GA*CAST*	126.09 ** (*p* = 0.008)	173.58 ± 0.37 ^c^	36
*LEP*	A69G	23 A Synonymous	Barki	28	GB1*LEP*	58.84 ** (*p* = 0.006)	414.03 ± 0.47 ^b^	74
	-	-		32	GB2*LEP*	67.25 ** (*p* = 0.006)	421.25 ± 0.72 ^a^	78
	C89A, G305A	30 A to D Non-synonymous 102 R to Q Non-synonymous	Aboudeleik	41	GA1*LEP*	86.17 ** (*p* = 0.005)	178.32 ± 0.61 ^c^	37
	G186A	62 L Synonymous		10	GA2*LEP*	21.01 ** (*p* = 0.006)	170.07 ± 0.39 ^d^	35
	-	-		9	GA3*LEP*	18.91 ** (*p* = 0.006)	172.35 ± 0.54 ^d^	36
*MYLK4*	A94G	32 T to A Non-synonymous	Barki	35	GB1*MYLK4*	73.56 ** (*p* = 0.006)	422.30 ± 0.47 ^a^	79
	C172T	58 R to C Non-synonymous		8	GB2*MYLK4*	16.81 ** (*p* = 0.005)	415.40 ± 0.27 ^b^	75
	-	-		17	GB3*MYLK4*	35.72 ** (*p* = 0.009)	415.22 ± 0.56 ^b^	74
	C44A C249T	15 A to D Non-synonymous 83 S Synonymous	Aboudeleik	44	GA1*MYLK4*	92.47 ** (*p* = 0.005)	175.79 ± 0.26 ^c^	37
	-			16	GA2*MYLK4*	33.62 ** (*p* = 0.009)	171.37 ± 0.35 ^d^	34
*MEF2B*	C55T	19 R to C Non-synonymous	Barki	37	GB1*MEF2B*	77.76 ** (*p* = 0.006)	420.18 ± 0.53 ^a^	77
	-	-		23	GB2*MEF2B*	48.34 ** (*p* = 0.009)	415.10 ± 0.42 ^b^	75
	G157A C297G	53 G to R Non-synonymous 99G Synonymous	Aboudeleik	38	GA1*MEF2B*	79.86 ** (*p* = 0.004)	175.31 ± 0.29 ^c^	37
	-	-		22	GA2*MEF2B*	46.24 ** (*p* = 0.006)	171.85 ± 0.29 ^d^	35
*STAT5A*	A134G	45 Q to R Non-synonymous	Barki	18	GB1*STAT5A*	37.83 ** (*p* = 0.008)	415.53 ± 0.17 ^b^	75
	-	-		42	GB2*STAT5A*	88.27 ** (*p* = 0.004)	419.75 ± 0.48 ^a^	77
	-	-	Aboudeleik	60	GA*STAT5A*	126.09 ** (*p* = 0.002)	173.58 ± 0.22 ^c^	36
*TRPV1*	A235G	79 K to E Non-synonymous 145 T to M Non-synonymous	Barki	17	GB1*TRPV1*	35.72 ** (*p*= 0.006)	420.16 ± 0.32 ^a^	77
	C434T			20	GB2*TRPV1*	42.03 ** (*p* = 0.005)	418.87 ± 0.66 ^b^	80
	A235G, C434T			9	GB3*TRPV1*	18.91 ** (*p* = 0.009)	415.71 ± 0.41 ^c^	74
	-			14	GB5*TRPV1*	29.42 ** (*p* = 0.009)	415.82 ± 0.39 ^c^	73
	-	-	Aboudeleik	60	GA*TRPV1*	126.09 ** (*p* = 0.009)	173.58 ± 0.37 ^d^	36

*CAST* = calpastatin; *LEP* = leptin; *MYLK4* = myosin light chain kinase family member 4; *MEF2B* = myocyte enhancer factor 2B; *STAT5A* = signal transducer and activator of transcription 5A; *TRPV1* = transient receptor potential cation channel subfamily V member 1. A = Alanine; C = Cisteine; D = Aspartic Acid; E = Glutamic Acid; G = Glycine; K = Lysine; L = Leucine; M = Methionine; Q = Glutamine; R = Argnine; S = Serine; T = Threonine; and W = Tryptophan. Means within the same column of, different litters are significantly different at (*p* < 0.01). ** = Significant at (*p* < 0.01).

## Data Availability

The data that support the findings of this study are available from the corresponding author upon reasonable request.

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
