# Peer review of "Analysis of Potential Genes and Economic Parameters Associated with Growth and Heat Tolerance in Sheep (*Ovis aries*)"

_animals, 2023, doi:10.3390/ani13030353_

Round 1
Reviewer 1 Report
The authors explored the potential genes and economic factors that might associated with growth and heat tolerance in two sheep breeds. The manuscript was not well designed and organized. Besides, the expression of this manuscript was too poor, and needed to be improved greatly. Accordingly, I suggest reject of this manuscript.
My comments are as follows:
1. The main limitation of the study is that the hypothesis and methods are not well establish. It is difficult to follow the reasoning of the authors, and how they reached the conclusions. The manuscript needs more details about the experimental design and the analyses performed.
2. Only 120 sheep (60 for Aboudeleik vs. 60 for Barki) are sampled in this study. However, for an animal experiment, it is not sufficient to make convincing results.
3. The FI and FCR of each sheep should be calculated, so as to be more accurate. Is there a big difference between individuals of the same breed?
4. Line 151: Explain why those genes were chosen.
5. The qulity of all figures need to be improved and please modified the fomat of tables.
6. Can economic benefits be correlated with different SNP loci?
Author Response
Comments and Suggestions for Authors
The authors explored the potential genes and economic factors that might associated with growth and heat tolerance in two sheep breeds. The manuscript was not well designed and organized. Besides, the expression of this manuscript was too poor, and needed to be improved greatly. Accordingly, I suggest reject of this manuscript.
My comments are as follows:
Comment
- The main limitation of the study is that the hypothesis and methods are not well establish. It is difficult to follow the reasoning of the authors, and how they reached the conclusions. The manuscript needs more details about the experimental design and the analyses performed.
Response
We thank reviewer for this comment. Our manuscript contains highlights and novelty statements that recommend it for publication as follows:
- Previous research examined the growth performance of sheep based on phenotypic characteristics. However, no studies have previously examined sheep growth traits from an economic standpoint and by taking into account a candidate gene.
- Many studies on heat stress in livestock have primarily focused on temperature and relative humidity. The molecular basis of sheep heat tolerance is not well understood. Therefore, it is necessary to do additional research involving functional verification by using genetic approaches that better incorporate computational and statistical analyses.
- When compared to the corresponding GenBank reference sequence, it is interesting to note that the polymorphisms found and published here reveal new information about the studied genes.
Comment
- Only 120 sheep (60 for Aboudeleik vs. 60 for Barki) are sampled in this study. However, for an animal experiment, it is not sufficient to make convincing results.
Response
We thank reviewer for this comment. Previous studies used the same number and obtained convincing results in ISI ranked journals as follows:
- Ahmed EL-SAYED, Maged EL-ASHKER, Eman EBISSY, Ahmed ATEYA (2020). EFFECT OF PREPARTUM VITAMIN E and SELENIUM ADMINISTRATION ON POSTPARTUM GENE EXPRESSION and METABOLIC PROFILE OF IMMUNE AND OXIDATIVE MARKERS IN BARKI EWES. Genetika, 52 (2), 673-688.
- Ateya, A., El-Sayed, A. & Mohamed, R. Gene expression and serum profile of antioxidant markers discriminate periparturient period time in dromedary camels. Mamm Res 66, 603–613 (2021). https://doi.org/10.1007/s13364-021-00578-3.
- Ateya A.I., El-Seady Y.Y., Atwa S.M., Merghani B.H., Sayed N.A. Novel single nucleotide polymorphisms in lactoferrin gene and their association with mastitis susceptibility in Holstein cattle. Genetika, 2016 48(1):199-210.
- Darwish A, Ebissy E, Ateya A, El-Sayed A. Single nucleotide polymorphisms, gene expression and serum profile of immune and antioxidant markers associated with postpartum disorders susceptibility in Barki sheep. Anim Biotechnol. 2021 Aug 18:1-13. doi: 10.1080/10495398.2021.1964984. Epub ahead of print. PMID: 34406916.
- Mona Al-Sharif, Hend Radwan, Basma Hendam, Ahmed Ateya (2022). DNA polymorphisms of FGFBP1, leptin, k-casein, and as1-casein genes and their association with reproductive performance in dromedary she-camels. Theriogenology, 178 (2022) 18-29.
- Mona M. Al-Sharif, Hend A. Radwan, Basma M. Hendam, Ahmed I. Ateya, Exploring single nucleotide polymorphisms in GH, IGF-I, MC4R and DGAT1 genes as predictors for growth performance in dromedary camel using multiple linear regression analysis, Small Ruminant Research, Volume 207, 2022, 106619.
Comment
- The FI and FCR of each sheep should be calculated, so as to be more accurate. Is there a big difference between individuals of the same breed?
Response
We thank reviewer for this comment. FI and FCR are calculated as means for each breed as there is no big difference between individuals of the same breed.
Comment
- Line 151: Explain why those genes were chosen.
Response
- We thank reviewer for this comment. Previous studies expanded on the relationship between sheep growth features and polymorphisms in the CAST, LEP, MEF2B, and STAT5A genes. However, the latter studies used PCR-RFLP approach for exploring the association of investigated genes with growth traits. We investigated this association with the SNP genetic marker, in contrast to earlier investigations.
- In discussion section we mentioned the role of each investigated gene in growth and heat tolerance in domestic animals.
- Some of investigated genes are scarcely reported to be associated with growth and heat tolerance in ruminants. Additionally there were no previous literature on some of investigated genes in sheep. Therefore we tried to investigate their relationship with growth and heat tolerance by aid of PubMed published sheep sequence and designing of new primers.
- Noteworthy mentioning that, the identified SNPs are first identified when compared with those found in GenBank.
Comment
- The quality of all figures need to be improved and please modified the format of tables.
Response
We thank reviewer for this comment. Noteworthy mentioning that the figures are submitted as supplementary data; however the quality of figures is improved. The format of tables is also modified.
Comment
- Can economic benefits be correlated with different SNP loci?
Response
We are grateful to the reviewer for drawing it to our consideration. We make a correlation between economic benefits with different SNP loci. Due to nucleotide sequence variants in growth related genes, the growth performance as a phenotype will differ that will affect economic parameters as costs.
Reviewer 2 Report
The authors have made a good attempt to conduct a sound study. However, I have a major concern regarding some parts of the objectives in this study. One part, assessing the economic parameters between the two breeds, screening the genetic variance for growth performance related genes and their association study, is appreciable. However the other portion “understand the genetic variance in heat adaptability in the two breeds, another goal was to examine molecular characterization of heat tolerance genes” as stated between lines 102-104 which is also another objective of the study does not seem to be addressed appropriately.
For the growth performance/economic traits, the authors were able to associate the SNPs/gene tic variations in the growth performance related genes with their relevant traits. However this was not done for the heat tolerance genes.
The authors have not considered any of the heat stress/thermal tolerance related traits (physiological/morphological/blood biochemical/endocrine, etc) to associate the genetic variation between animals to their adaptability/tolerance/resilience. Hence mere detection of SNPs in these genes cannot be claimed to indicate the animal/breed to be tolerant to heat stress. Hence it would be better that the authors sort this issue. Following would be my suggestion to tackle this issue:
1. Improvise the statistical model:
The temperature-humidity index is a standard and crucial variable used by researchers globally when working on heat stress/climate change studies in livestock. If the authors are able to gain access to the weather variables (mainly temperature and relative humidity) during the study period, it will be possible to calculate the THI. This factor (fixed factor) can then be included in the model.
2. If the authors had recorded any of the earlier mentioned heat stress/thermal tolerance related traits, then incorporate them as an additional trait (dependent variable, ‘y’ in the model) that can be used for the genetic association study
3. Refrain from making any claims regarding heat tolerance in the entire study.
The authors could also come forward with other alternatives however, this concern needs to be addressed to ensure the scientific soundness of the work.
Author Response
Reviewer 2
Comment
The authors have made a good attempt to conduct a sound study. However, I have a major concern regarding some parts of the objectives in this study. One part, assessing the economic parameters between the two breeds, screening the genetic variance for growth performance related genes and their association study, is appreciable. However the other portion “understand the genetic variance in heat adaptability in the two breeds, another goal was to examine molecular characterization of heat tolerance genes” as stated between lines 102-104 which is also another objective of the study does not seem to be addressed appropriately.
Response
We are grateful to the reviewer for drawing it to our consideration. “Understand the genetic variance in heat adaptability in the two breeds, another goal was to examine molecular characterization of heat tolerance genes” which is also another objective of the study is addressed appropriately in materials, results and discussion sections.
Comment
For the growth performance/economic traits, the authors were able to associate the SNPs/gene tic variations in the growth performance related genes with their relevant traits. However this was not done for the heat tolerance genes.
Response
We are grateful to the reviewer for drawing it to our consideration. The association is done in results and fully deciphered in discussion section.
Comment
The authors have not considered any of the heat stress/thermal tolerance related traits (physiological/morphological/blood biochemical/endocrine, etc) to associate the genetic variation between animals to their adaptability/tolerance/resilience. Hence mere detection of SNPs in these genes cannot be claimed to indicate the animal/breed to be tolerant to heat stress. Hence it would be better that the authors sort this issue. Following would be my suggestion to tackle this issue:
- Improvise the statistical model:
The temperature-humidity index is a standard and crucial variable used by researchers globally when working on heat stress/climate change studies in livestock. If the authors are able to gain access to the weather variables (mainly temperature and relative humidity) during the study period, it will be possible to calculate the THI. This factor (fixed factor) can then be included in the model.
- If the authors had recorded any of the earlier mentioned heat stress/thermal tolerance related traits, then incorporate them as an additional trait (dependent variable, ‘y’ in the model) that can be used for the genetic association study
- Refrain from making any claims regarding heat tolerance in the entire study.
The authors could also come forward with other alternatives however, this concern needs to be addressed to ensure the scientific soundness of the work.
Response
We are grateful to the reviewer for drawing it to our consideration. All suggested inquiries are made. Therefore we have added surplus parts in in materials, results and discussion sections.
Reviewer 3 Report
An interesting and well-written research paper. This paper is useful and acceptable.
Please see some minor comments/suggestions.
1. Please provide the hypothesis for this study
2. L 301 spelling mistake...”both”
3. It would be very interesting to see a hypothetical figure or a summarized version of your findings as an illustration (if possible)
4. L404 Please be consistent on acronyms wherever you have used it (examples, HSP90 or Hsp90)
5. Is there any supplementary files/data that you could provide for the economic evaluation parameters? (if not, it would be interesting to see a value for these parameters
Author Response
Reviewer 3
Comments and Suggestions for Authors
Comment
An interesting and well-written research paper. This paper is useful and acceptable.
Please see some minor comments/suggestions.
Response
We thank reviewer for this positive comment. In fact our paper contains highlights and novelty statements that recommend it for publication.
Comment
- Please provide the hypothesis for this study Potential genes and economic parameters associated with growth and heat tolerance in sheep
Response
We thank reviewer for this comment. We added null and alternative hypothesis in the statistical analysis section as follows:
H0: Genetic polymorphisms and economic parameters are not associated with growth and heat tolerance in Barki and Aboudeleik sheep.
HA: Genetic polymorphisms and economic parameters are associated with growth and heat tolerance in Barki and Aboudeleik sheep.
Comment
- L 301 spelling mistake...”both”
Response
We thank reviewer for this comment. The spelling mistake is corrected.
Comment
- It would be very interesting to see a hypothetical figure or a summarized version of your findings as an illustration (if possible)
Response
We thank reviewer for this comment. We have already submitted a graphical abstract for our study including design and our findings. However, we will resubmit it again upon your inquiry.
Comment
- L404 Please be consistent on acronyms wherever you have used it (examples, HSP90 or Hsp90)
Response
We thank reviewer for this comment. It is corrected
Comment
- Is there any supplementary files/data that you could provide for the economic evaluation parameters? (if not, it would be interesting to see a value for these parameters
Response
We thank reviewer for this comment. As a novelty, we make a correlation between economic benefits with different SNP loci. Due to nucleotide sequence variants in growth related genes, the growth performance as a phenotype will differ that will affect economic parameters as net returns and economic efficiency.
Round 2
Reviewer 2 Report
The authors have made a considerable effort to improvise the manuscript.
1. There are some basic formatting issues that need to be looked into:
a. Line spacing and text format for the contents inserted between line numbers 194-215
b. Also the subheading 2.8 needs to be aligned
c. Line 353- typographic error ‘physic-chemical’ which should be ‘physico-chemical’
d. Line 420- typing error ‘(P< 0.011)’?
Kindly do a thorough spell-check and format-check throughout the manuscript
Author Response
Dear Editor,
Thank you very much for your interest and consideration. It is of great pleasure to receive your decision on our paper "Potential genes and economic parameters associated with growth and heat tolerance in sheep (Ovis aries)"
Please find our revised manuscript and the response to the reviewer comments. All comments were considered while revising this paper.
We hope that, our response is satisfactory to both editor and reviewers.
Thank you very much for your cooperation and waiting for your response
Kind regards
Sincerely yours,
Corresponding author
Reviewer 1
Comments and Suggestions for Authors
The authors explored the potential genes and economic factors that might associated with growth and heat tolerance in two sheep breeds. The manuscript was not well designed and organized. Besides, the expression of this manuscript was too poor, and needed to be improved greatly. Accordingly, I suggest reject of this manuscript.
Comments and Suggestions for Authors
The authors have made a considerable effort to improvise the manuscript.
- There are some basic formatting issues that need to be looked into:
Comment
- Line spacing and text format for the contents inserted between line numbers 194-215
Response
We thank reviewer for this comment. Line spacing and text format for the contents inserted between line numbers 194-215 are reformatted.
- Also the subheading 2.8 needs to be aligned
Response
We thank reviewer for this comment. Subheading 2.8 are aligned.
- Line 353- typographic error ‘physic-chemical’ which should be ‘physico-chemical’
Response
We are grateful to the reviewer for drawing it to our consideration. Typographic error ‘physic-chemical’ changed into ‘physico-chemical throughout the manuscript.
- Line 420- typing error ‘(P< 0.011)’?
Response
We are grateful to the reviewer for drawing it to our consideration. Typing error ‘(P< 0.011) is corrected.